# Seed Transmission of Three Viruses in Two Pear Rootstock Species *Pyrus betulifolia* and *P. calleryana*

**DOI:** 10.3390/v14030599

**Published:** 2022-03-14

**Authors:** Liu Li, Lihong Wen, Guoping Wang, Yuzhuo LYU, Zuokun Yang, Xiaoping Yang, Qingyu Li, Ni Hong

**Affiliations:** 1Key Lab of Plant Pathology of Hubei Province, College of Plant Science and Technology, Huazhong Agricultural University, Wuhan 430070, China; 15032210591@163.com (L.L.); wenlihong05@163.com (L.W.); gpwang@mail.hzau.edu.cn (G.W.); yuzhuolv@163.com (Y.L.); 13297974203@163.com (Z.Y.); 2Key Laboratory of Horticultural Crop (Fruit Trees) Biology and Germplasm Creation of the Ministry of Agriculture, Wuhan 430070, China; 3National Sand Pear Germplasm Repository in Wuchang, Research Institute of Fruit and Tea, Hubei Academy of Agricultural Science, Wuhan 430064, China; yangxiaoping1981@163.com; 4Yantai Academy of Agricultural Science, Yantai 264000, China; liqingyu613891@163.com

**Keywords:** apple chlorotic leaf spot virus, apple stem grooving virus, apple stem pitting virus, *P. betulifolia*, *P. calleryana*, seed transmission, sequence

## Abstract

Viral seed transmission causes the spread of many plant viral diseases. *Pyrus*
*betulifolia* and *P. calleryana* are important rootstock germplasms for pear production in China. This study revealed the widespread infection of apple stem grooving virus (ASGV), apple chlorotic leaf spot virus (ACLSV), and apple stem pitting virus (ASPV) in maternal trees of *P. betulifolia* and *P. calleryana* by nested multiplex reverse transcription-polymerase chain reaction (nmRT-PCR) assays. Seeds from eight *P. betulifolia* and two *P. calleryana* trees had positive rates of 15.9–73.9%, 0–21.2%, and 40.4% for ASGV, ASPV, and ACLSV, respectively. At the cotyledon and 6–8 true leaf stages, seedlings grown from seeds of infected trees gave positive rates of 5.4% and 9.3% for ASGV, 6.7% and 15.6% for ACLSV, and 0% and 2.7% for ASPV, respectively. Incidence in nursery *P. betulifolia* seedlings of 10.1%, 5.3%, and 3.5% were determined for ASGV, ACLSV, and ASPV, respectively. The nucleotide sequences of coat protein (CP) and movement protein coding genes of both ASGV and ASPV, and CP gene of ACLSV from maternal trees, seeds, and seedlings were analyzed. Sequence identities and phylogenetic comparison with corresponding sequences from GenBank demonstrated that molecular variation occurred within ASGV, ACLSV, and ASPV isolates, with most sequences determined here had close relationships with reported isolates infecting pear or formed independent clades. This is the first report on the seed transmission and the molecular characteristics of these viruses infecting two rootstock species. These findings provided important evidence in management effort for pear viral diseases.

## 1. Introduction

Viruses are important pathogens of threat to crop production. Pear (*Pyrus* spp.) and apple (*Malus* spp.) are the most widely grown pome fruit crops. *Apple stem grooving virus*, *Apple chlorotic leaf spot virus* and *Apple stem pitting virus* are the type species of the genera *Capillovirus*, *Trichovirus*, and *Foveavirus* in the family *Betaflexiviridae* in the order *Tymovirales* [1], respectively. Each of these viruses possess a single-stranded positive-sense genomic RNA with a poly (A) tail at 3’-terminus. Apple stem grooving virus (ASGV), apple chlorotic leaf spot virus (ACLSV), and apple stem pitting virus (ASPV) commonly occur in commercially cultivated pear and apple trees worldwide [2]. ASGV also infects other important woody fruit crops, including species in genera *Prunus*, *Actinidia*, *Citrus*, and some ornamental plants. ACLSV also infects *Prunus* spp. and causes pseudo-pox disease of peach [3]. There are also reports of ASPV infecting cherry and sour cherry in India [4], and *Cydonia japonica*, *P. calleryana*, and *P. amygdaliformis* in Greece [5]. Most pear and apple trees infected by these viruses are asymptomatic, but their growth can be reduced by virus infection [2,6,7]. However, in some pear varieties, ACLSV infection may induce severe leaf malformation and chlorotic rings or line patterns [2,8]. ASGV is found to be associated with pear black necrotic leaf spot (PBNLS) in Korea [9] and ASPV infection causes vein yellowing or red mottling [7,10,11] and fruit stony pits on susceptible pear trees [12] and is also associated with quince (*C. oblonga*) fruit deformation disease [5,13]. ASGV and ASPV induce xylem pits in the stem of *M. pumila* ‘*Virginia crab*’ [2,14,15]. The mixed infections of the three viruses are very common in pear and apple plants, and can cause top-working disease of pear and apple trees on susceptible rootstocks, resulting in significant reduction in fruit quality and yield [7,8,15,16,17,18,19,20,21]. The severity of symptoms elicited by those viruses depends largely on the plant species and virus strains [8,17,18]. Our recent study showed that the co-infection of ASGV and ASPV significantly decreased the growth, and proliferation ability of in vitro cultivated plants of *P. communis* cv. ‘*Conference*’, and ASGV infection strongly inhibited the root development of in vitro cultivated plants of *P. pyrifolia* cv. ‘Jinshui no. 2′ [22].

Pear is a perennial fruit crop, in which viruses can replicate and spread to the whole plant, and then persistently affect the plant development during its whole life. Pear viruses are commonly disseminated through the movement and utilization of virus-infected propagative materials and plants [2,16]. Grafting virus-infected scions are the major ways of virus distribution among individuals. Until now, there have been no surveys to determine whether the three viruses can naturally spread among host pear and apple trees in orchards. Although the mechanical transmission of the three viruses is effective from infected saps of their natural hosts, including pear and apple, to some herbaceous host plants [1,2], there is no evidence for spread by direct contact between healthy and infected woody host plants. The natural insect vector of these viruses has not been determined yet. Then, certified virus-tested propagation materials have been used for efficient management of viral diseases of apple and pear trees worldwide [18,19,23,24].

To obtain virus-free propagative materials of pear and apple, great efforts have been exerted to develop reliable techniques for the detection and elimination of viruses [21,25,26,27,28,29,30]. Seedlings of *P. betulifolia* and *P. calleryana* are widely-used rootstocks for pear tree propagation in China. In previous studies, we established a nmRT-PCR method for the efficient detection of the three viruses ASGV, ASPV, and ACLSV widely infecting pear trees and found that some seedlings of *P. betulifolia* and *P. calleryana* were positive for these viruses [27], which caused our attention for the possible seed transmission of these viruses. If the rootstocks of pear are contaminated with seed-borne viruses, the propagated pear plants can become infected by viruses. In view of the importance of the three viruses and the risk of virus infection in rootstocks for the production of certified pear germplasms and plants, the current study was conducted to identify virus-infected maternal trees of *P. betulaefolia* and *P. calleryana*, evaluate the localizations of the three viruses in flora tissues, the transmission rate through seeds of *P. betulaefolia* and *P. calleryana*, and molecular characteristics of the three viruses from the two rootstock species. The results provide important information for the understanding of pear virus dispersion and the management of these viruses, avoiding introduction of these viruses into new certified pear germplasms through virus-infected rootstocks.

## 2. Materials and Methods

### 2.1. Plant Materials

To test whether fruiting trees of *P. betulifolia* and *P. calleryana* were infected by ASGV, ACLSV and ASPV, leaf samples were collected from 31 trees of *P. betulifolia* and three trees of *P. calleryana* grown in Shandong, Shanxi, Gansu and Hubei provinces and tested for the presence of ASGV, ACLSV and ASPV.

For the validation of seed borne and seed transmission frequencies of the three viruses, open-pollinated seeds were collected from fruits of eight *P. betulifolia* and two *P. calleryana* trees, which were positive for one or more of ASGV, ACLSV, and ASPV. All seeds were fully washed under tap water and air-dried, then stored at 4 °C for 3–6 months. Partial seeds were used as samples for testing seed borne transmission of the three viruses, and some other seeds were sown in potted-soil, then grown in a green house. Cotyledons of germinated seeds and the six true leaves at 6–8 leaf stage of the seedlings were assayed for viral seed transmission.

To validate the incidence of the three viruses in rootstock seedlings, leaf samples were randomly collected from 821 nursery-seedlings grown from commercial seeds of *P. betulifolia*.

### 2.2. RNA Extraction and RT-PCR

Considering the low titers of ASGV, ACLSV, and ASPV in rootstock samples, the multiple nested RT-PCR (nmRT-PCR) assay previously developed was used for the simultaneous detection of the three viruses [27]. Total RNA from pear plant samples was extracted according to a rapid silica spin column-based method described recently [31]. In the nmRT-PCR assay, cDNA synthesis was performed at 37 °C for 1 h by using a universal primer M4-T [GTTTTCCCAGTCACGAC(T)_15_] to target the poly (A) tail of these viruses, and M-MLV reverse transcriptase (Promega, Madison, WI, USA). The primer pairs, PCR reaction solution and thermal cycling condition were the same as those reported previously [27].

The complete coat protein (CP) and movement protein (MP) genes of the three viruses were amplified by nested RT-PCR (nRT-PCR) reactions. The cDNA synthesis was performed as mentioned above. For the first round of amplification, the universal primer M4-R was combined with primers ASGV-MP-F, ACLSV-CP-F, and ASPV-90-110-F specifically targeting to ASGV, ACLSV, and ASPV sequences. For the second cycle of PCR, primer pairs SG-MPF/R and SG-CPF/R were used for the amplifications of MP and CP genes of ASGV, primer pair CL-CPF/R was used for the amplification of CP gene of ACLSV, and primer pairs TGB-F/R and 90-110-F/1500-1520-R were used to amplify whole TGB and CP genes of ASPV. The sequences and targeting sites of primers used for the amplification were listed in Appendix A.

In all RT-PCR tests, in vitro cultured pear (*P. communis*) plants infected with ASGV, ACLSV, and ASPV were used as positive controls, and leaf samples from a virus-free seedling of *P. betulifolia* were used as negative controls [21]. The PCR products of nmRT-PCR and nRT-PCR assays were electrophoresed on 2% and 1.2% agarose gels in TAE buffer, respectively, stained with ethidium bromide (0.5 µg/mL) and visualized under UV illumination.

### 2.3. RNA Sequencing and Amplification of ASPV Genomic RNA

One seedling of *P. betulifolia* (ID: DL) was subjected to RNA sequencing (RNA-Seq) analysis. The total RNA extraction, and the generation and sequencing of the cDNA library of rRNA-depleted RNAs were performed by Biomarker Biology Technology Ltd. Company (Beijing, China) as reported previously [32]. Briefly, ribosomal RNA (rRNA) in total RNA extract was removed using an Epicentre Ribo-ZeroTM rRNA removal kit (Epicentre, Madison, WI, USA). The prepared rRNA-depleted RNA sample was used to construct a cDNA library with a TruSeq RNA Sample Prep Kit v2 (Illumina, San Diego, CA, USA) and subjected to high throughput sequencing on an Illumina HiSeq XTen sequencing machine (Illumina, San Diego, CA, USA) with a paired-end 150 bp setup. The raw RNA reads were quality-filtered by removing adapter sequences and low-quality reads with more than 5% Ns or with 20% base quality values (Q20) less than 20 using FASTP version 1.5.6 [33]. Then, the obtained clean RNA reads were de novo assembled into contigs using IDBA-UD version 1.1.1 [34] with k-mer values of 80, 90, and 110. Contigs were subsequently screened for virus sequence identities against the NCBI databases (http://www.ncbi.nlm.nih.gov/ (accessed on 17 August 2020)) using BlastX and BlastN programs. After identification of ASPV in the sample, primers were designed based on the viral contig sequences (Appendix A) and the gaps between contigs were filled by RT-PCR amplifications (Figure 1A). The viral reads were measured and mapped to virus genome by Samtools version 1.5 [35]. The read coverage was calculated using genomecov package in BEDtools version 2.27.1 [36] and visualized using CIRCOS [37].

### 2.4. Gene Cloning and Sequence Analysis

PCR products were gel-purified and ligated into the pMD18-T vector (Takara, Dalian, China). Positive clones identified by PCR assays were sequenced at Shanghai Sangon Biological Engineering and Technology and Service Co. Ltd., Shanghai, China. Phylogenetic analysis was performed by aligning the obtained sequences with corresponding sequences referred from GenBank using Muscle method in MEGA 7.0 [38]. Phylogenetic trees were constructed using the maximum-likelihood method based on the Tamura-Nei model with 1000 bootstrap replicates. To have a clear view of the relationships between the newly obtained sequences and those determined from different hosts of each virus, the referred sequences were selected basing on their similarities, phylogenetic positions, and host origins.

## 3. Results

### 3.1. Maternal Trees of P. betulaefolia and P. calleryana Positive for Three Viruses

Initial nmRT-PCR tests showed that ASGV had the highest incidence, with 27 out of 34 samples (accounting for 79.4%) were positive for ASGV (Table 1). Of these samples, 11 and three samples (accounting for 36.7% and 9.1%) were positive for ASPV and ACLSV, respectively. The mixed infection of ASGV and ASPV was found in six and four trees of *P. betulifolia* from Shanxi and Shandong provinces. Of three *P. calleryana* trees, one was positive for ASGV and another one was positive for ASGV and ACLSV, respectively.

### 3.2. Presence of Three Viruses in Different Tissues of P. betulaefolia and P. calleryana

After the initial detection of viruses in *P. betulaefolia* and *P. calleryana* trees, 17 *P. betulaefolia* and two *P. calleryana* trees were selected to investigate virus distribution within infected trees. The phloem, leaf, flower, petal, and pollen samples were individually collected from these trees and tested for ASGV, ACLSV, and ASPV by using nmRT-PCR. The three viruses could be detected in all these tissue samples, but the detection efficiency differed in different tissues (Table 2). For ASGV and ACLSV, the leaf and flower samples had relatively high positive frequency, followed by petal and pollen samples, and phloem samples showed lower positive frequency. However, for ASPV, the virus was positive in five phloem samples, negative in other tissue samples from the same trees. Among ASGV positive trees as tested using leaf samples, other tissue samples of one tree (SX-Pb38) were negative, and petal and pollen samples of SX-Pb6 were negative. These results demonstrated that these viruses were present not only in leaves, but also in flowers of some plants, and also further confirmed the presence of one or more viruses in these plants.

### 3.3. Seed Borne of P. betulaefolia and P. calleryana for Three Viruses

In order to understand the incidence of the three viruses in seeds of *P. betulaefolia* and *P. calleryana*, seeds were collected from eight *P. betulaefolia* trees (SX-Pb16, SX-Pb17, SX-Pb23, SX-Pb27, SX-Pb40, SD-Pb1, HB-Pb1, HB-Pb2) and two *P. calleryana* trees (HB-Pc1 and HB-Pc2) located in Shanxi (SX), Hubei (HB), and Shandong (SD) provinces and individually tested for the three viruses (Table 3). Among 385 seeds from *P. betulaefolia*, the frequencies of ASGV-positive seeds varied from 15.9% to 73.9% according to its host sources, and out of 232 seeds from four ASPV-positive trees, 0–21.2% seeds were positive for ASPV. Notably, the maternal trees SX-Pb40 and SD-Pb1 were ASPV negative, but one and two seeds from the two trees were ASPV-positive. Similarly, SX-Pb27 and SD-Pb1 were negative for ACLSV, but one seed from each of the trees was ACLSV-positive. Out of 54 seeds from a *P. calleryana* tree HB-Pc1 (positive for ASGV), eight seeds (accounting for 14.8% of tested seeds) were ASGV-positive. From HB-Pc2 (positive for ASGV and ACLSV), initially, 52 seeds were tested for the three viruses, the frequencies of ASGV-and ACLSV-positive seeds were 30.8% and 40.4%, respectively. Surprisingly, ten seeds (accounting for 19.2%) of HB-Pc2 were ASPV-positive. However, when seeds from HB-Pc2 were stored at 4 °C for 14 months, only three out of 91 seeds were positive for ASGV, and ASPV, and ACLSV were negative in these seeds. Mixed infections of ASGV/ACLSV/ASPV, ASGV/ASPV, ASGV/ACLSV, and ACLSV/ASPV were detected in one, 11, five, and four seeds (Appendix A), respectively.

### 3.4. Seed Transmission and Incidence of Three Viruses in Seedlings of P. betulaefolia and P. calleryan

According to the above detection results for three viruses, we confirmed that two maternal trees (HB-Pb1 and HB-Pb2) of *P. betulaefolia* were positive for ASGV and ASPV, a *P. calleryana* tree HB-Pc1 was positive for ASGV, and a *P. calleryana* tree HB-Pc2 was positive for ASGV and ACLSV (Table 4). In order to analyze the seed transmission of the three viruses to seedlings, seeds harvested from these virus positive trees were sowed in pots. The cotyledons and the sixth true leaves taken from seedlings at the cotyledon stage and 6–8 leaf stage were tested for the three viruses by nmRT-PCR. Generally, seed transmission rates for each virus fluctuated depending on host sources and their growth stages. Of 112 cotyledon and 205 true leaf samples from four ASGV-infected maternal trees, 6 (5.4%) and 19 (9.3%) gave positive results for the virus, with positive rates of 13.0%, 4.6%, 0%, and 4.4% at cotyledon stage, and 19.7%, 4.4%, 6.3%, and 2.2% at 6–8 leaf stage in HB-Pb1, HB-Pb2, HB-Pc1, and HB-Pc2 seedlings, respectively. Of 45 cotyledon and 45 true leaf samples from an ACLSV-infected maternal tree HB-Pc2, 3 (6.7%) and 7 (15.6%) gave positive results for ACLSV. Of 112 true leaf samples from two ASPV-infected maternal trees, 3 (2.7%) gave positive results for the virus, but cotyledons from HB-Pb1 and HB-Pb2 were negative for ASPV. In addition, ASPV was detected in two out of 45 cotyledons of ASPV-negative Pc2, but not detected in true leaves of seedlings. Results showed that the three viruses could be transmitted through seeds, indicated that these viruses most likely moved from infested seed to emerging seedlings.

To understand the infection statues of the three viruses in field-grown *P. betulaefolia* seedlings, a total of 877 seedlings of *P. betulaefolia* were collected randomly from nursery fields, and individually tested for the presence of ASGV, ACLSV, and ASPV by nmRT-PCR. Totally, 77, 39, and 28 seedlings (accounting for 8.8%, 4.5%, and 3.2%) were positive for ASGV, ACLSV, and ASPV. Mixed infection of two viruses was detected in 12 plants, of which five plants were positive for both ASGV and ACLSV, six plants were positive for both ASGV and ASPV, and one plant was positive for both ACLSV and ASPV (Appendix A).

### 3.5. Identification of Viruses Infecting P. betulaefolia Using High Throughput Sequencing

From the RNA-Seq analysis, a total of 67,420,486 clean reads were obtained from the sample DL. The resulting contigs from de novo assembling were subjected to BlastX and BlastN searches against the nr and nt databases at the NCBI. From the sample DL, six contigs with lengths ranging from 392 to 2695 bp matched the sequences of ASPV genome (Figure 1A and Appendix A). Except for the 392 bp contig (c9611) showing a 97.7% nt identity with an ASPV accession no. KU798310, other five contigs showed the highest identity of about 97% with the corresponding sequences of an ASPV isolate LYC (accession no. 763895) from *P. bretschneideri* cv. ‘chili’ specifically grown at Laiyang area in China [39]. The results indicated the presence of ASPV in the sample DL. We did not identify any contig in the sample homologous to other plant viruses. To have an overview of the molecular feather of ASPV from DL, the virus genome was determined by overlapping sequences of RT-PCR generated amplicons. Here, the virus isolate was named as ASPV-DL (accession no. OM313385). The genome of ASPV-DL consisted of 9269 nucleotides (nt) and shared 97.7% nt sequence identity with ASPV isolate LYC. Analyses of RNA reads deriving from the sample DL revealed 886 reads (accounting for 0.001314% of total clean RNA reads) mapped to the genome of ASPV-DL with a coverage rate of 98% (Figure 1B). However, the highest read depth was only near 50, thus un-resembling the distribution profiles of virus-derived RNAs reported for plus-sense RNA viruses [40].

### 3.6. Genetic Diversity and Phylogenetic Analysis of ASGV from P. betulaefolia and P. calleryana

The complete CP gene of ASGV was amplified from seven maternal trees, five seeds and seven seedlings of *P. betulifolia*, and from one maternal tree and two seedlings of *P. calleryana*, respectively. The complete MP gene of ASGV was amplified from nine maternal trees and one seed of *P. betulifolia*, and one maternal tree of *P. calleryana*, respectively. The sequences were submitted to GenBank with an accession numbers OM313332-OM313355 for CP clones and OM313356-OM313366 for MP clones. CP sequences of these ASGV isolates shared 87.3–100% nt and 94.1–100% aa similarities with each other, and 82.6–99.9% nt and 93.1–100% aa similarities with the corresponding sequences of ASGV isolates available at NCBI GenBank. The MP sequences of ASGV isolates shared 84.0–99.9% nt and 94.4–100% aa similarities with each other, and 86.2–100% nt and 90.7–100% aa similarities with the corresponding sequences of ASGV isolates available at NCBI GenBank. It was noticed that the clones of CP gene from a maternal tree HB-Pb1, and three seeds (HB-Pb1-18Z, HB-Pb1-6Z, and HB-Pb1-3Z), one seedling (HB-Pb1-7S) derived from the same maternal tree had diversity about 14.5% at nt level and 14.0% at aa level. Similarly, clones from either HB-Pc2 or SD-Pb34 also showed diversity of 12.13% at nt level and 4.64% at aa level. Among the obtained MP clones, two clones (HB-Pb1-6Z and HB-Pb1-22Z) from a seed of the maternal tree HB-Pb1 showed 85.5% nt and 86.0% aa similarity with each other. These results indicated sequence variation within ASGV isolates.

When the nucleotide sequences of ASGV CP and MP genes obtained in this study and the represent sequences referred from GenBank were subjected to phylogenetic analyzes, sequences from Pb and Pc distributed in different branches consisting of sequences from different hosts (Figure 2), indicating that the molecular composition of ASGV might be not related to their host origins. Five CP sequences from a fruiting tree HB-Pb1, its seeds and seedlings distributed into two branches, and two clones from HB-Pc2, and three clones from SD-Pb34 also in separated clades. In the MP-based tree, clones from Pb and Pc also clustered in different clades separated from other reported sequences. The result further confirmed molecular diversity within ASGV population from Pb and Pc.

### 3.7. Genetic Diversity and Phylogenetic Analysis of ACLSV from P. betulaefolia and P. calleryana

The complete CP gene of ACLSV was amplified from one maternal tree HB-Pc2 of *P. calleryana* and one seed (HB-Pc2-16) derived from the tree, and nine seedlings (SD-Pb134, SD-Pb28, SD-Pb17, SD-Pb75, SD-Pb40, SD-Pb143, SD-Pb144, SD-Pb51, and SD-Pb39) of *P. betulifolia,* respectively. The sequences were submitted to GenBank with an accession numbers OM313321-OM313331. The CP gene clones of these ACLSV isolates shared 88.8–100% nt and 93.8–100% aa sequence similarities, and 68.4–100% nt and 72–100% aa sequence similarities with the corresponding sequences of ACLSV isolates available at NCBI GenBank. It was noticed that the nucleotide sequences of CP clones HB-Pc2 and HB-Pc2-16 differed by 11.2%. Phylogenetic analysis of CP nucleotide sequences of ACLSV isolates showed that the CP clones determined from Pb and Pc clustered into one group, in which, except for HB-Pc-2 in the separated clade consisting sequences from apple and pear, all Pb and Pc in the clade consisting of pear isolates (Figure 3A).

Multiple alignment of ACLSV CP sequences obtained from this study and representative sequences referred from GenBank revealed uneven distribution of the variances along the CP and that the amino acid combination S40-M/V59-Y/C75-T130-L184, similar to the S40-L59-Y75-T130-L184 identified by Yaegashi et al. (2007) [41], presented in clones analyzed in this study (Figure 3B), suggesting that these clones were belonged to B6 type.

### 3.8. Genetic Diversity and Phylogenetic Analysis of ASPV from P. betulaefolia and P. calleryana

The complete CP gene of ASPV was amplified from seven *P. betulifolia* trees, one seed (HB-Pb2-7) derived from a *P. betulifolia* fruiting tree (HB-Pb2) and one seed (HB-Pc2-52) from a *P. calleryana* fruiting tree (HB-Pc2), respectively. The sequences were submitted to GenBank with an accession numbers OM313367-OM313377. Sequence analysis showed that the CP gene sequences of ASPV from these samples were highly variable and shared 65.3–99.8% nt and 70.2–99.7% aa identities with each other, and 64.7–99.8% nt and 70.9–99.7% aa identities with the corresponding sequences available at NCBI GenBank. Multiple alignment of ASPV CP protein sequences obtained from this study revealed variations in sizes with 410, 405, 403, 399, 398, and 394 aa. However, these sequences were in the range of ASPV CP sizes. The complete triple gene block protein coding genes (TGB1, TGB2, and TGB3) (accession nos. OM313378-OM313384) of ASPV were amplified from seven *P. betulifolia* trees. The TGB1, TGB2, and TGB3 genes shared 76–99.7%, 77.1–99.4%, and 85–99.5% nt identities, and 87–100%, 74.7–99.7%, and 84.3–100% aa identities with each other, and 75.3–99.6% nt and 86.5–100%, 72.5–100%, and 67.1–100% aa identities with the corresponding sequences available at NCBI GenBank, respectively. Two clones SX-Pb6-1 and SX-Pb6-8 from a *P. betulifolia* tree had high sequence diversity, with 73% nt and 81.2% aa identity.

In the CP- and TGB-based trees, ASPV sequences from *P. betulifolia* and *P. calleryana* distributed in different groups, and in each group Pb and Pc sequences showed close relationship with isolates from pear except for SX-Pb6-8 close to a hawthorn isolate (Figure 4). Inter sample variants presented in HB-Pb2 and SX-Pb6, with three CP clones of HB-Pb2 separately distributed in three groups, and two CP clones of SX-Pb6 in two clades. It was notable that two clones SX-Pb36-2 and LN-Pb1 were closely related to isolate LYC characterized from a Chinese local pear cultivar [39]. When the nucleotide sequences of TGB1, TGB2 and TGB3 were separately analyzed (Appendix A), the newly obtained sequences fitted into the same groups as those derived from the conjugated TGB sequences.

## 4. Discussion

This study revealed the natural infection of three viruses ASGV, ASPV and ACLSV in *P. betulifolia* and *P. calleryana* trees. The ASGV infection frequency in tested *P. betulaefolia* was about 79%, which is close to the incidence of the virus in cultivated pear trees [42,43], whilst ASPV and ACLSV had have relatively lower incidence. These trees naturally grown in fields, and we do not know whether they suffered grafting or other artificial manipulation, then their infection sources are not clear. Although ASGV was not detected in samples of *P. calleryana*, we could not exclude its infection in *P. calleryana* since limited samples were tested. Pear leaves are frequently used as samples for virus detection. In this study, different tissues of *P. betulaefolia* and *P. calleryana* were tested for the three viruses. ASGV and ACLSV were efficiently detected in leaf samples of *P. betulaefolia* and *P. calleryana*. However, ASPV had a relatively high positive rate in phloem samples. The results revealed uneven distribution of the virus in these plants and virus concentrations below the detection limits in some samples might result in negative detection results [44]. In whole flower, petal and pollen samples of 17 *P. betulaefolia* and two *P. calleryana* trees, the three viruses were detectable, indicating these viruses could infect the reproductive organs.

Seed transmission plays a pivotal role in the spread of plant viruses and viroids [45,46,47,48]. This work demonstrated that the seed-borne of three viruses commonly infecting pear. The seed-borne frequencies fluctuated among virus species and host sources. We found that the detectable rates of ASGV decreased greatly from 30.8% to 3.3% and ACLSV was not detectable in *P. calleryana* seeds after a storage for 14 months. The diverse effects of storage on titer of different viruses might be resulted from their different stabilities. Among the three pear viruses, ASGV is mostly stable to thermal therapy [25]. A previous study showed that storage for seven months, and heat-treatment of seeds at temperatures near the thermal inactivation point of TRSV, failed to inactivate the virus in infected seeds [45,49,50]. We noticed that two and three seeds from negative material trees of *P. betulaefolia* were ASPV- and ACLSV-positive, respectively, and even 11 seeds from a negative material tree of *P. calleryana* were ASPV-positive. Some fruit tree viruses such as prunus necrotic ringspot virus (PNRSV) and raspberry bushy dwarf virus (RBDV) can infect trees via virus-carrying pollen [51,52]. The pollen-borne transmission of ASPV and ACLSV might cause seed contamination. Continued investigation of the role of pollen-borne viruses in seed infection is required to further reveal seed transmission mechanism of these viruses.

Our results showed that the three viruses could be transmitted through seeds with transmission rates varying greatly by virus species and material trees. The seed transmission rates of the three viruses in pear rootstock maternal trees to the emerging seedlings fluctuated from 2% to 19%, which was much lower than seed-borne rates. The inactivation of viruses during seed storage and germination might reduce their seed transmission rates. The accumulation and genotypes of viruses play an important role in the transmission [53,54]. The percentages of virus positive seedlings at 6–8 leaf stage was relatively higher than that of seedlings at cotyledon stage, which might be caused by relatively low virus titers at cotyledon stage and the accumulated levels of these viruses in seedlings at 6–8 leaf stage. ASGV had relatively higher seed transmission rates of 0–13.0% in the cotyledons and 2.2–19.7% to the six true leaves. ASPV in two *P. betulaefolia* trees was not detectable in cotyledons and showed transmission rates of 3.03% and 2.17% in seedlings at 6–8 leaf stage. ACLSV in a *P. calleryana* tree had transmission rates of 6.7% in cotyledons, and 15.6% in the sixth true leaves. We noticed that ASPV was detected in two out of 45 cotyledons of ASPV-negative Pc2, but not detected in true leaf samples of its seedlings. Some stable viruses, such as tobamoviruses, are carried as a contaminant on the seed coat, and cause seedling infection by mechanical transmission during the germination and first stages of growth [46,55]. Then, seed disinfection treatments were used to reduce virus infection [55,56,57,58]. Although all seeds tested in this study were washed extensively under tap water during harvesting, we still could not exclude the contamination possibility of viruses on seeds, which possibly caused the occurrence of positive reactions of a few seeds and cotyledons from negative maternal trees. The low titers and seed borne rates of these viruses made it difficult to localize these viruses in seeds. Considering the difficulty of mechanical transmission of these viruses among natural hosts, the infection is not likely to occur through the micro-lesions caused during the germination and initial growth stages. Then, our results suggest that the embryo infection could be responsible for the transmission of the three viruses from infected seeds to seedlings. Further investigation confirmed the infection of the three viruses in field-grown seedlings of *P. betulaefolia*, with positive rates of 10.1%, 5.3%, and 3.5% for ASGV, ACLSV, and ASPV, which were in the ranges of seed transmission rates from infected trees. Importantly, mixed infections of two viruses in field-grown seedlings of *P. betulaefolia*, which might cause the mixed infections occurring in pear plant propagated on these seedlings. RNA-seq analysis also confirmed the infection of ASPV in a *P. betulaefolia* seedling sample. The low RNA read depth covering the viral genome indicated a low virus titer in the sample, coincided with the low detection efficiency in conventional RT-PCR [27]. Virus infection in rootstocks can result in substantial incidence on pear trees, which raises the need for an increased caution for virus infection during virus-free germplasm propagation and conservation. As used for some seed transmission viruses [55,56,57,58], seed disinfection treatments might be employed to reduce virus infection in pear rootstocks.

Pear plants are commonly propagated by grafting on rootstocks. The long cultivation history with wide and multiple artificial manipulations of infected scion and rootstocks might contribute to the wide and mixed infection of pear viruses. Previous studies mainly focused on the molecular characteristics of the three viruses commonly infecting pear and apple and indicated that the phylogenetic groupings of ASGV, ACLSV, and ASPV might be correlated to their host origins [43,59,60,61,62,63,64]. This study for the first time revealed molecular diversity within each of the three viruses from *P. calleryana* and *P. betulaefolia*. The sequence variation range of each virus from the two rootstock species was similar to that from pear trees cultivated in China [43,60,61,62]. The CP and MP or TGB sequences of ASGV and ASPV isolates determined from *P. calleryana* and *P. betulaefolia* distributed in distinct phylogenetic groups, indicating that molecular variation occurred within the virus populations from pear rootstocks. However, in each group of ASGV and ASPV, sequences from *P. calleryana* and *P. betulaefolia* had relatively close phylogenetic relationship with isolates from cultivated pear plants, suggesting similar host origins for isolates of both ASGV and ASPV infecting rootstocks and cultivated pear plants, which might also contribute the common infection of these viruses in pear trees. All CP sequences of ACLSV determined here grouped into one clade represented by the isolate JB from a Chinese pear tree [64] and some other isolates from apple and hawthorn, but were distal to the reported clade IV containing isolates from sandy pear [61], suggesting that ACLSV isolates from sandy pear and rootstock might have different origins. Mixed infection of molecular variants of each virus presented in maternal rootstock trees and/or their seeds. However, we do not know whether it is caused by unknown grafting manipulations a long time ago or via virus-carrying pollen as some other fruit tree viruses [51], or even other vectors. In an un-controlled environment, molecular variation of virus genome might also occur due to variable selection pressures, which could cause the difference of viral sequences from maternal pear trees and seeds. However, the current phylogenetic relationships between virus isolates from rootstocks and those from cultivated pear trees are insufficient to explain the origins of these viruses, which needs further extensive investigation for population composition of these viruses from different natural hosts and their natural dismission routes.

To our knowledge, this is the first report that demonstrates the seed transmission and the molecular composition of three viruses infecting two pear rootstock germplasms. Seeds play important roles in disseminating plant virus diseases [65,66]. These findings are especially important for avoid virus dissemination during the propagation of virus-free pear germplasms and plants. Thus, identifying virus-free mother plants for rootstock seed production can help reduce damage caused by these viruses.

## Figures and Tables

**Figure 1 viruses-14-00599-f001:**
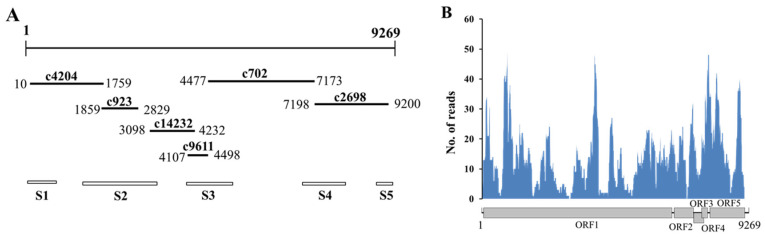
RNA-Seq derived contigs (**A**) and profile distribution of RNA reads (**B**) along genomic RNA of ASPV isolate DL. (**A**) Black bold lines represent contigs and their positions at the genome of ASPV-DL were indicated. Rectangles indicate the five segments (S) amplified by nRT-PCR. (**B**) A schematic diagram of the genome organization of ASPV isolate DL was used as reference.

**Figure 2 viruses-14-00599-f002:**
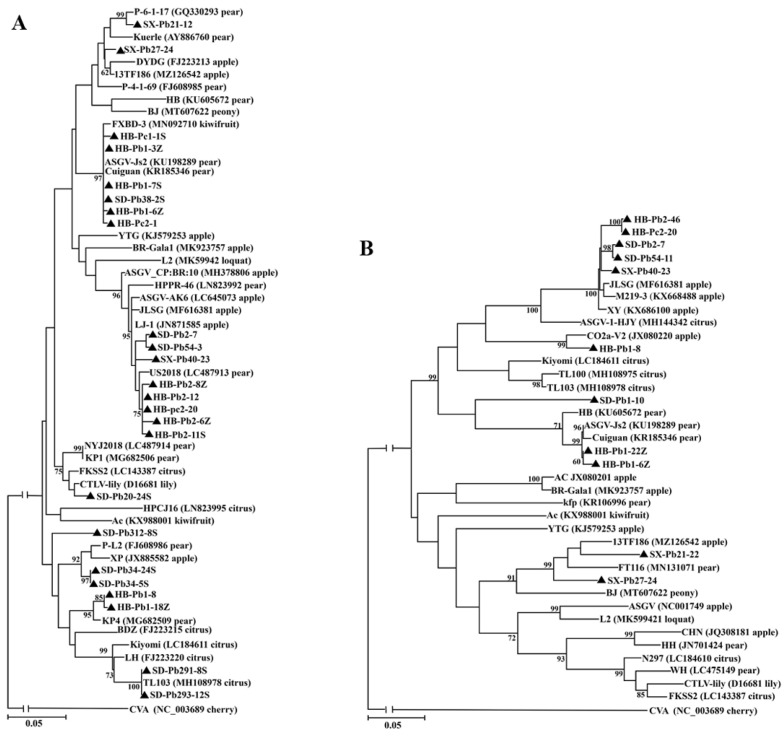
Unrooted maximum-likelihood (ML) phylogenetic trees generated from the nucleotide sequences of CP (**A**) and MP (**B**) genes of apple stem grooving virus (ASGV). The reported sequences in each tree were marked by its isolate name followed by a GenBank accession number and host name. The sequences determined in this study were highlighted by black triangles and marked by their geographic origins, hosts and clone IDs, and sequences from seedlings and seeds were identified with ‘S’ and ‘Z’. The corresponding RNA sequence of cherry virus A (CVA) was used as a outgroup in each phylogenetic tree.

**Figure 3 viruses-14-00599-f003:**
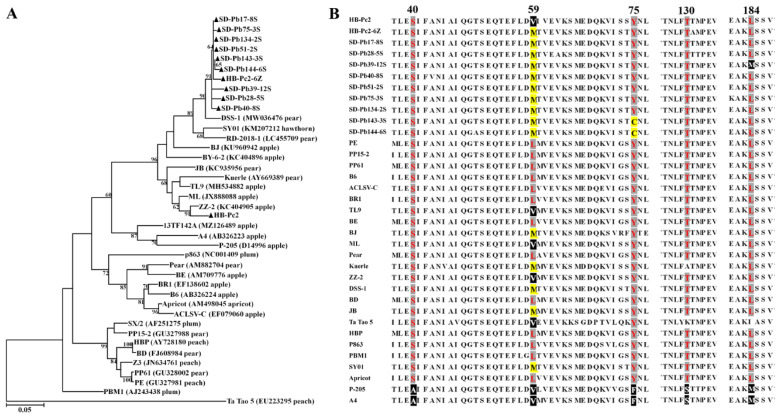
Unrooted maximum-likelihood (ML) phylogenetic tree generated from the nucleotide sequences of CP gene (**A**) and multiple amino acid sequence alignment of CP (**B**) of apple chlorotic leaf spot virus (ACLSV). (**A**) Each of reported sequences in the tree was marked by its isolate name followed by a GenBank accession number and host name, the sequences determined in this study were highlighted by black triangles and marked by its geographic origin, host and clone ID, and sequences from seedlings and seeds were identified with ‘S’ and ‘Z’. The corresponding sequence of an isolate Ta tao 5 was used as a outgroup in the phylogenetic tree. (**B**) The amino acids at five positions 40, 59, 75, 130, and 184 for the discrimination of B6 and P-205 types were shaded with gray and black background colors, respectively, and yellow background color showed amino acids different from B6 type.

**Figure 4 viruses-14-00599-f004:**
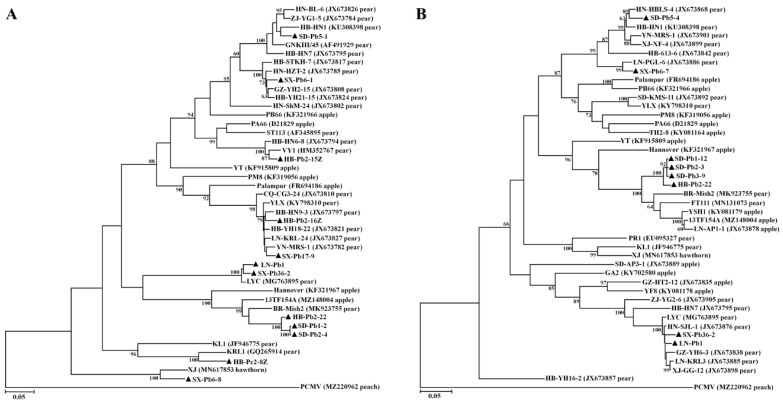
Unrooted maximum-likelihood (ML) phylogenetic trees generated from the nucleotide sequences of CP (**A**) and TGB (**B**) genes of apple stem pitting virus (ASPV). Each reported sequence was marked by its isolate name followed by a GenBank accession number and host name. The sequences determined in this study were highlighted by black triangles and marked by their geographic origins, hosts and clone IDs, and sequences from seeds were identified with ‘Z’. The corresponding sequence of peach chlorotic mottle virus (PCMV) was used as a outgroup in each phylogenetic tree.

**Table 1 viruses-14-00599-t001:** The infection statues of three viruses in maternal trees of *P. betulaefolia* and *P. calleryana* from different locations.

Species	Origin (Province)	Sample No.	Infected/%		
			ASGV	ACLSV	ASPV
*P. betulifolia*	Shanxi	10	10/100	2/20	6/60
	Shandong	5	5/100	0	4/80
	Gansu	12	7/58.3	0	0
	Hubei	4	3/75	0	1/25
*P. calleryana*	Hubei	3	2/66.7	1/33.3	0
Total		34	27/79.4	3/8.8	11/32.4

**Table 2 viruses-14-00599-t002:** The detection of three viruses in the different tissues of *P. betulaefolia* and *P. calleryana*.

Tree ID ^a^	Phloem			Leaf			Flower			Petal			Pollen		
	SG	CLS	SP	SG	CLS	SP	SG	CLS	SP	SG	CLS	SP	SG	CLS	SP
SX-Pb6	+	−	−	+	−	−	+	−	+	−	−	−	−	−	−
SX-Pb16	−	−	+	+	−	−	+	−	−	+	−	−	+	−	−
SX-Pb17	−	−	+	−	−	−	+	−	+	+	−	−	+	−	−
SX-Pb21	−	−	+	+	−	−	+	−	−	+	−	−	+	−	−
SX-Pb23	+	−	−	+	−	−	−	−	−	+	−	−	+	−	−
SX-Pb26	−	−	−	N	N	N	+	−	−	+	−	−	+	−	−
SX-Pb27	+	−	−	+	−	−	+	−	−	−	−	−	+	−	−
SX-Pb36	−	−	−	−	+	−	−	−	+	+	−	−	−	−	−
SX-Pb38	−	−	+	+	+	−	−	−	−	−	+	−	−	−	−
SX-Pb40	+	−	−	+	−	−	+	−	−	+	−	−	+	−	−
SD-Pb1	N	N	N	+	−	−	+	−	−	+	−	−	+	−	−
SD-Pb2	N	N	N	+	−	+	+	−	+	+	−	−	+	−	+
SD-Pb3	N	N	N	+	−	−	+	−	+	+	−	−	+	−	−
SD-Pb4	N	N	N	+	−	+	+	−	−	+	−	+	+	−	−
SD-Pb5	N	N	N	+	−	+	+	−	−	+	−	−	+	−	+
HB-Pb1	+	−	+	+	−	−	+	−	−	+	−	−	+	−	−
HB-Pb2	+	−	+	+	−	+	+	−	+	+	−	+	+	−	+
HB-Pc1	+	−	−	−	−	−	−	−	−	−	−	−	−	−	−
HB-Pc2	+	+	−	+	+	−	+	+		+	+	−	+	−	−

+: positive; −: negative; N: not tested; SG: apple stem grooving virus (ASGV); CLS: apple chlorotic leaf spot virus (ACLSV); SP: apple stem pitting (ASPV). ^a^ For tree IDs, SX, SD, and HB indicate that samples from Shanxi, Shandong, and Hubei provinces, and Pb and Pc indicate *P. betulaefolia* and *P. calleryana*, respectively.

**Table 3 viruses-14-00599-t003:** The seed borne frequencies of three viruses infecting *P. betulaefolia* and *P. calleryana*.

Maternal Tree				No. of Seeds	Positive/%		
	ASGV	ACLSV	ASPV		ASGV	ACLSV	ASPV
SX-Pb16	+	−	+	62	18/29	0	3/4.8
SX-Pb17	+	−	+	63	10/15.9	0	0
SX-Pb23	+	−	−	38	26/68.4	0	0
SX-Pb27	+	−	−	23	17/73.9	1/4.4	0
SX-Pb40	+	−	−	46	15/32.6	0	1/2.2
SD-Pb1	+	−	−	46	32/69.6	1/2.2	2/4.4
HB-Pb1	+	−	+	52	13/25	0	11/21.2
HB-Pb2	+	−	+	55	23/41.8	0	2/3.6
HB-Pc1	+	−	−	54	8/14.8	0	0
HB-Pc2	+	+	−	52	16/30.8	21/40.4	10/19.2
				91 ^a^	3/3.3	0	0

+: positive. −: negative. ^a^ Seeds stored for 14 months at 4 °C.

**Table 4 viruses-14-00599-t004:** The detection results of three viruses in seedings of *P. betulaefolia* and *P. calleryana*.

Maternal Tree	Maternal Tree			Cotyledon				True Leaf			
				No. of Plants	Infected/%			No. of Plants	Infected/%		
	SG ^a^	CLS	SP		SG	CLS	SP		SG	CLS	SP
HB-Pb1	+	−	+	23	3/13	0	0	66	13/19.7	0	2/3.03
HB-Pb2	+	−	+	22	1/4.6	0	0	46	2/4.4	0	1/2.17
HB-Pc1	+	−	−	22	0	0	0	48	3/6.3	0	0
HB-Pc2	+	+	−	45	2/4.4	3/6.7	2/4.4	45	1/2.2	7/15.6	0

^a^ SG: apple stem grooving virus (ASGV); CLS: apple chlorotic leaf spot virus (ACLSV); SP: apple stem pitting (ASPV).

## Data Availability

The data presented in this study are available in article and Appendix A.

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
