# Peer review of "Seed Transmission of Three Viruses in Two Pear Rootstock Species Pyrus betulifolia and P. calleryana"

_viruses, 2022, doi:10.3390/v14030599_

Round 1
Reviewer 1 Report
This paper presented interesting work on seed transmission of common apple viruses in pears. I don't think this has been extensively studied before so presents new work which is of value not only to the pear industry but the plant virus community as well. The paper is well written and scientifically sound. Only a few grammatical errors to fix (see highlighted comments in attached pdf) Well done.

Author Response
Response to Reviewer 1 Comments
Point1: This paper presented interesting work on seed transmission of common apple viruses in pears. I don't think this has been extensively studied before so presents new work which is of value not only to the pear industry but the plant virus community as well. The paper is well written and scientifically sound. Only a few grammatical errors to fix (see highlighted comments in attached pdf) Well done.
Response: We thank the reviewer for the evaluation. We have revised the manuscript according to the comments from the reviewer. All suggestions have also been accepted.
Point 2: Line 27 add (MP) after movement protein to keep consistent
Response: In abstract, movement protein appeared only once, so that we did not write its abbreviation. The abbreviations MP and TGBps for movement protein and triple gene block proteins were used in the MS text.
Reviewer 2 Report
In this study, the seed transmission and the molecular composition of three viruses infecting two pear rootstock germplasms were detected. The result had contribution to the pear virus prevention and treatment.
However, the manuscript is not so well prepared, many minor writings error appeared. Such as the Latin name for the P. betulifolia and P. calleryana, no full names were showed in the manuscript. At the line 179 of page 4, the word size is significant as normal, please check the writing completely.
And the pear growth in the field, for the seed transmission virus had no consistent aa identities, whether other similar virus might be transmitted by the insect? This case should discuss. As the condition is no in a control environment the virus strain can not be keep as single inducer.
Author Response
Response to Reviewer 2 Comments
Point1: In this study, the seed transmission and the molecular composition of three viruses infecting two pear rootstock germplasms were detected. The result had contribution to the pear virus prevention and treatment.
Response: We thank the reviewer for the evaluation and positive comments.
Point 2: However, the manuscript is not so well prepared, many minor writings error appeared. Such as the Latin name for the P. betulifolia and P. calleryana, no full names were showed in the manuscript. At the line 179 of page 4, the word size is significant as normal, please check the writing completely.
Response: We thank the reviewer for the kind corrections. We have checked all text and corrected grammatical errors. The full names Pyrus betulifolia in the abstract was used now. In the text, the genus name Pyrus appeared in the line 39, so that we used P. here after.
Point 3: And the pear growth in the field, for the seed transmission virus had no consistent aa identities, whether other similar virus might be transmitted by the insect? This case should discuss. As the condition is no in a control environment the virus strain can not be keep as single inducer.
Response: We agree with the review, many factors can cause the variation of viruses. In the revised MS, it was discussed.